# Revealing roles of S-layer protein (SlpA) in *Clostridioides difficile* pathogenicity by generating the first *slpA* gene deletion mutant

Shaohua Wang,[1,2] Maria C. Courreges,[1] Lingjun Xu,[3] Bijay Gurung,[1] Mark Berryman,[1] Tingyue Gu[3]

**ABSTRACT** *Clostridioides difficile* infection (CDI) with high morbidity and high mortality is an urgent threat to public health, and *C. difficile* pathogenesis studies are eagerly required for CDI therapy. The major surface layer protein, SlpA, was supposed to play a key role in *C. difficile* pathogenesis; however, a lack of isogenic *slpA* mutants has greatly hampered analysis of SlpA functions. In this study, the whole *slpA* gene was successfully deleted for the first time via CRISPR-Cas9 system. Deletion of *slpA* in *C. difficile* resulted in smaller, smother-edged colonies, shorter bacterial cell size, and aggregation in suspension. For life cycle, the mutant demonstrated lower growth (changes of optical density at 600 nm, OD600) but higher cell density (colony-forming unit, CFU), decreased toxins production, and inhibited sporulation. Moreover, the mutant was more impaired in motility, more sensitive to vancomycin and Triton X-100-induced autolysis, releasing more lactate dehydrogenase. In addition, SlpA deficiency led to robust biofilm formation but weak adhesion to human host cells.

**IMPORTANCE** *Clostridioides difficile* infection (CDI) has been the most common hospital-acquired infection, with a high rate of antibiotic resistance and recurrence incidences, become a debilitating public health threat. It is urgently needed to study *C. difficile* pathogenesis for developing efficient strategies as CDI therapy. SlpA was indicated to play a key role in *C. difficile* pathogenesis. However, analysis of SlpA functions was hampered due to lack of isogenic *slpA* mutants. Surprisingly, the first *slpA* deletion *C. difficile* strain was generated in this study via CRISPR-Cas9, further negating the previous thought about *slpA* being essential. Results in this study will provide direct proof for roles of SlpA in *C. difficile* pathogenesis, which will facilitate future investigations for new targets as vaccines, new therapeutic agents, and intervention strategies in combating CDI.

**KEYWORDS** *Clostridioides difficile*, S-layer protein, pathogenesis, biofilm, pathogen-host interaction

Address correspondence to Shaohua Wang, wangs4@ohio.edu.

The authors declare no conflict of interest.

See the funding table on p. 12.

*C*lostridioides difficile is a gram-positive, spore-forming obligate anaerobe. *C. difficile* infection (CDI) causes colitis and leads to more than 500,000 emergency visits and around 29,000 deaths each year in the USA alone (1), which incurs an estimated annual cost of $6.3 billion (2). The Center for Disease Control and Prevention has announced *C. difficile* as an urgent and life-threatening pathogen (3). Pathogenesis studies of *C. difficile* are urgently required for CDI therapy. One of the critical factors among *C. difficile* pathogenesis is the adhesion to host intestinal mucosa, which is supposed to be mediated by the paracrystalline surface layer (S-layer) of *C. difficile* (4, 5). The *C. difficile* major S-layer is primarily composed of two proteins, the high-molecular-weight and the low-molecular-weight S-layer proteins, namely, HMW SLP and LMW SLP. The two SLPs

are derived through post-translational cleavage of a precursor protein (SlpA) encoded by a single gene, *slpA* (6, 7). A cysteine protease, Cwp84, was demonstrated to play a role in the maturation of SlpA. However, it was also indicated as not an essential virulence factor since that *C. difficile* mutant with the deletion of *cwp84* was still able to cause disease (8). Researches using isolated SLPs demonstrated their inhibition of attachment of *C. difficile* to human epithelial cells (6). These reports have suggested the crucial role of SlpA in interactions with the host, but detailed analysis of the contributions of the SlpA to pathogenesis is still to be clarified.

Nonetheless, a lack of isogenic *slpA* mutants has greatly hampered analysis of *C. difficile* S-layer functions (9). The S-layer was initially thought to be essential, as evidenced by an inability to generate any transposon-mediated insertional mutants within the *slpA* gene (10). Interestingly, a premature frameshift mutation in *slpA* gene was obtained by chance (11), and the frameshift mutant is avirulent during infection, indicating the essential role of SlpA in *C. difficile* pathogenesis. To analyze essential genes, a CRISPR interference (CRISPRi) system with xylose-inducible dCas9 was developed (12). Based on this CRISPRi system, the expression of SlpA was knocked down, and the *slpA*-depleted cells demonstrated reduced sporulation and increased lysozyme sensitivity (12). However, no further studies were performed to illustrate the roles of SlpA in pathogenesis. Moreover, the limitations of CRISPRi, such as polarity on downstream or upstream genes (13), plus the instability of plasmid and the requirement for xylose induction make the CRISPRi system poorly fit for *in vivo* animal studies (12). Therefore, it is necessary to knock out the whole gene completely to analyze the roles of SlpA in *C. difficile* and in CDI pathogenesis.

Another serious threat factor contributing to CDI pathogenesis is biofilm formation. Biofilms possess much higher local sessile cell concentrations, and they can be far more recalcitrant than planktonic cells (14), and link to up to 80% of bacterial infections (15, 16). Biofilms play prominent roles in CDI recurrence and antibiotic resistance (17). Adherence is the first and most essential step of the biofilm growth cycle (18). As the major S-layer surface protein, SlpA has been implicated in *C. difficile* biofilm formation. However, whether the S-layer *per se* is involved in biofilm formation remains uncertain due to lack of mutant with deleted *slpA* gene. Disruption of Cwp84 resulted in uncleaved SlpA, but the effects of *cwp84* disruption on biofilm formation were dependent on the strain background. *Cwp84* mutants reduced biofilm formation in strain R20291 (19), while disruption of Cwp84 increased biofilm formation in strain 630 (20). Therefore, studies with *slpA* mutants are necessary for illustrating roles of SlpA in biofilm formation and recurrent CDI pathogenesis.

In this study, to challenge the possibility of deleting the whole *slpA* gene in *C. difficile* and provide direct proof for roles of SlpA in *C. difficile* pathogenesis, we generated the first *slpA* deletion mutant of *C. difficile* via CRISPR-Cas9 system. Based on this novel mutant, roles of SlpA in phenotype of *C. difficile*, steps of life cycle, sensitivity to antibiotics, Triton X-100-induced autolysis, motility, and biofilm formation of *C. difficile*, as well as in its adhesion to human epithelial cells were demonstrated. Results here will advance our understanding of the importance of SlpA to *C. difficile* and its roles in CDI pathogenesis, which will assist developing new targets as novel therapeutic approaches to combat CDI and recurrent CDI.

## RESULTS

### Generation of *slpA* deletion *C. difficile* mutant

pSlpA2 harboring CRISPR-Cas9 system, guiding sequence targeting on *slpA* gene, and homology arm sequences around *slpA* gene were constructed, transferred into *C. difficile*, and induced for screening mutants, followed by curing pSlpA2. In Fig. 1A, the shorter band verified the deletion of the whole *slpA* gene (2,160 bp) in *C. difficile* 630Δ*erm*Δ*slpA*. Moreover, S-layer proteins were extracted, and the loss of HMW and LMW SlpA bands in *C. difficile* 630Δ*erm*Δ*slpA* further confirmed removal of SlpA at protein level (Fig. 1B).

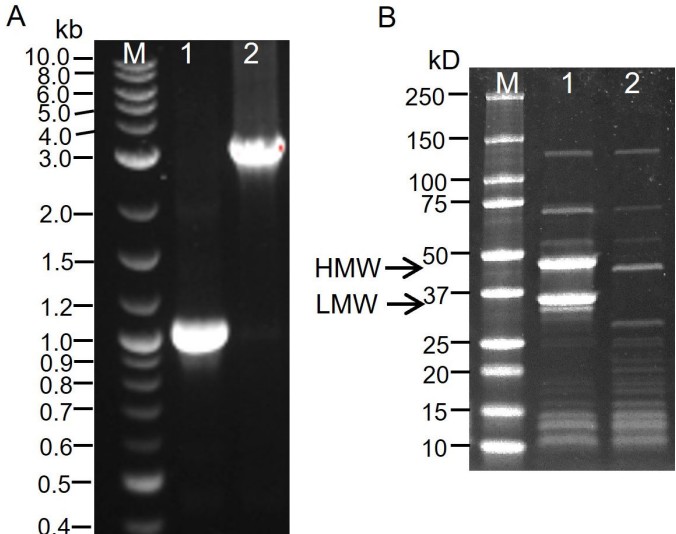

**FIG 1** Generation of *slpA* deletion mutant of *C. difficile*. (A) Colony PCR product (1,078 bp in lane 1) verified deletion of *slpA* in the mutant strain *C. difficile* 630Δ*erm*Δ*slpA*, compared with the 3,272 bp band (lane 2) amplified from the wild-type *C. difficile* 630Δ*erm*. (B) S-layer proteins were separated on SDS-PAGE gel. Compared to proteins from *C. difficile* 630Δ*erm* in lane 1, missing HMW and LMW bands of SlpA (lane 2) further confirmed deletion of *slpA* in *C. difficile* 630Δ*erm*Δ*slpA*. The NEB 1 kb DNA ladder (A) and the Bio-Rad precision plus protein standards (B) were used as markers (lane M) with numbers on the left representing the band length.

Along with disappearance of SlpA bands, other cell wall proteins including those around 50–75 kDa were lowly expressed, and those around 15–50 kDa were highly expressed.

## Roles of SlpA in morphology and life cycle of *C. difficile*

Compared to the wild-type *C. difficile* 630Δ*erm* with thick and rough-edged colonies, *C. difficile* 630Δ*erm*Δ*slpA* demonstrated smother-edged and translucent colonies which were only half size of that from the wild-type colonies (Fig. S1A). Bacterial length of *C. difficile* 630Δ*erm*Δ*slpA* observed under microscope was around 1 µm shorter than the wild type (Fig. S1B). Deletion of *slpA* also affected steps of the life cycle of *C. difficile*. Interestingly, although lower growth OD600 (60% of that from the wild type) (Fig. 2A) was detected with *C. difficile* 630Δ*erm*Δ*slpA* strains, cell density (colony numbers) at 12 h (Fig. 2B) and 24 h (Fig. S2A) was significantly higher in the mutant, then, decreased and became lower than the wild type at 48-h cultivation (Fig. S2A). Extracellular toxin production (Fig. 2C), intercellular toxin production (Fig. S2B), and sporulation (Fig. 2D) were all inhibited in *C. difficile* 630Δ*erm*Δ*slpA* due to deficiency of SlpA.

## Roles of SlpA in motility, antibiotic sensitivity, and autolysis of *C. difficile*

Without SlpA, motility of *C. difficile* 630Δ*erm*Δ*slpA* was impaired, demonstrating 50% swimming zone of that from the wild type (Fig. 3A). Moreover, *C. difficile* 630Δ*erm*Δ*slpA* exhibited a vancomycin minimum inhibitory concentration (MIC) of 0.8 µg/mL, slightly (0.2 µg/mL) less than that for *C. difficile* 630Δ*erm*, but *C. difficile* 630Δ*erm*Δ*slpA* was more susceptible to vancomycin at sub-MICs (Fig. 3B). Changes in bacterial surface architecture due to SlpA deficiency also affected Triton X-100-induced autolysis (Fig. 3C). *C. difficile* 630Δ*erm*Δ*slpA* exhibited a more rapid rate of Triton X-100-induced autolysis, with 75.7% lysed cells, compared with only 3.2% from the wild type in 4 h. Alone with its ease for Triton X-100-induced autolysis, more intercellular proteins such as lactate dehydrogenase (LDH) were released from *C. difficile* 630Δ*erm*Δ*slpA* (Fig. 3D).

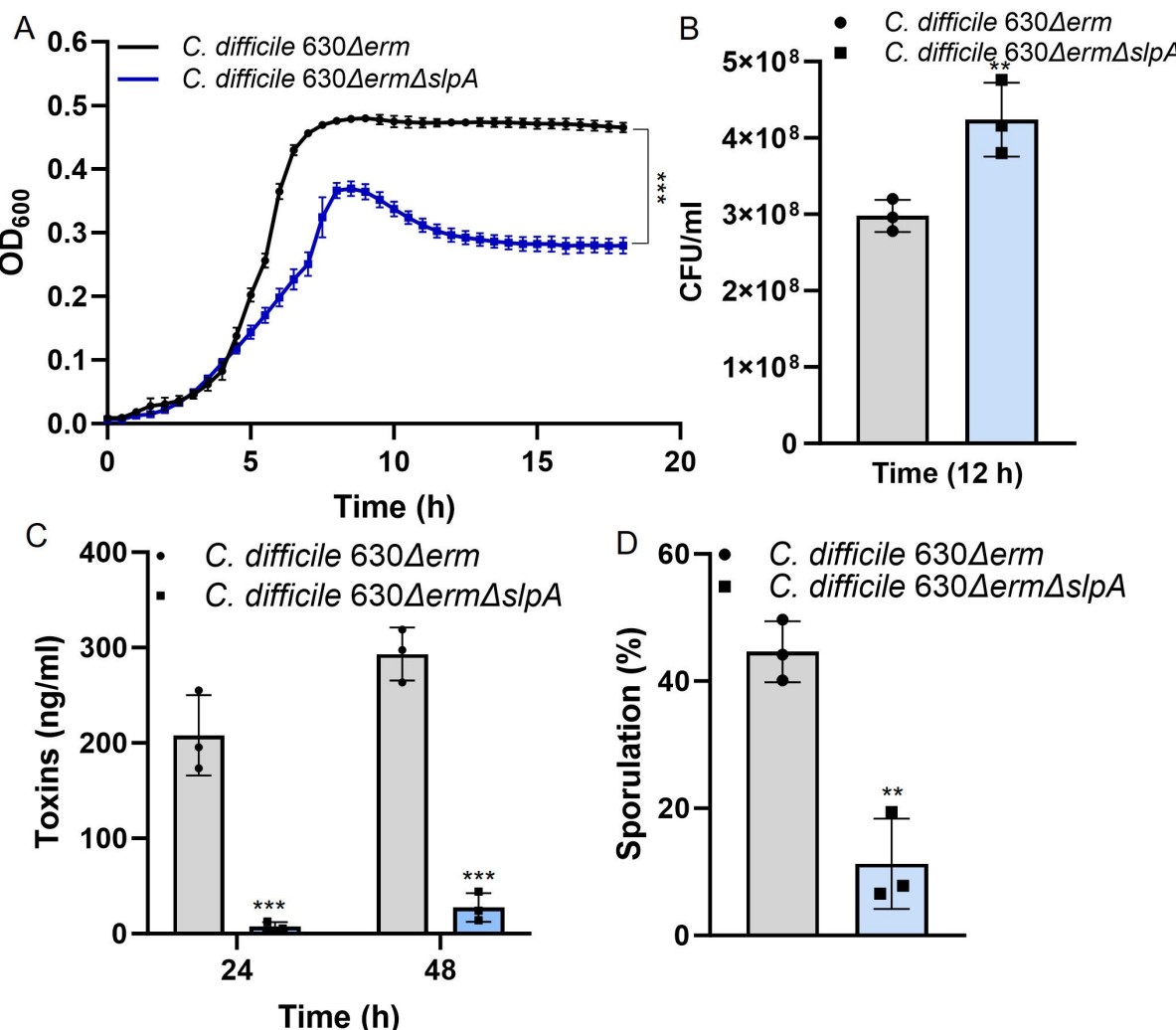

**FIG 2** Effects of SlpA on life cycle of *C. difficile*. *C. difficile* 630Δ*erm*Δ*slpA* deficient in SlpA production demonstrated lower growth OD (A) but higher colony numbers in cultures after 12-h cultivation (B), less toxin production determined with Toxin A & B enzyme-linked immunosorbent assay (ELISA) Kit (C), and limited sporulation (D). Growth curve was performed in six replicates, and other assays were carried out in triplicate; all assays were repeated three times. *P*-values for differences between the wild-type and mutant strains, **$P < 0.01$ and ***$P < 0.001$.

## Roles of SlpA in biofilm formation of *C. difficile*

*C. difficile* 630Δ*erm*Δ*slpA* demonstrated a propensity to aggregate at the bottom of the culture tubes (Fig. S2C), indicating potential biofilm formation ability changes in the mutant strain (17). As shown with crystal violet (CV) staining, the darker color of *C. difficile* 630Δ*erm*Δ*slpA* suggested increased biofilm formation due to SlpA deficiency (Fig. 4A). *C. difficile* biofilm possesses a slight corrosivity against 304 stainless steels, which allowed us to monitor its biofilm formation using an electrochemical cell (e-cell) in real time. Corrosion resistant ($R_p$) data in Fig. 4B suggested that the biofilms of both wild-type and the *slpA* deletion mutant became mature at 3 d incubation when a comparable stable low $R_p$ was reached. Because *C. difficile* 630Δ*erm*Δ*slpA* showed higher corrosivity during the 7-d incubation period, its biofilm was considered to have a consistently higher sessile cell count than the wild-type *C. difficile* 630Δ*erm* (21). Confocal laser scanning microscopy (CLSM) also demonstrated increased biofilm biomass in *C. difficile* 630Δ*erm*Δ*slpA* with stronger fluorescent signal intensity (Fig. 4C).

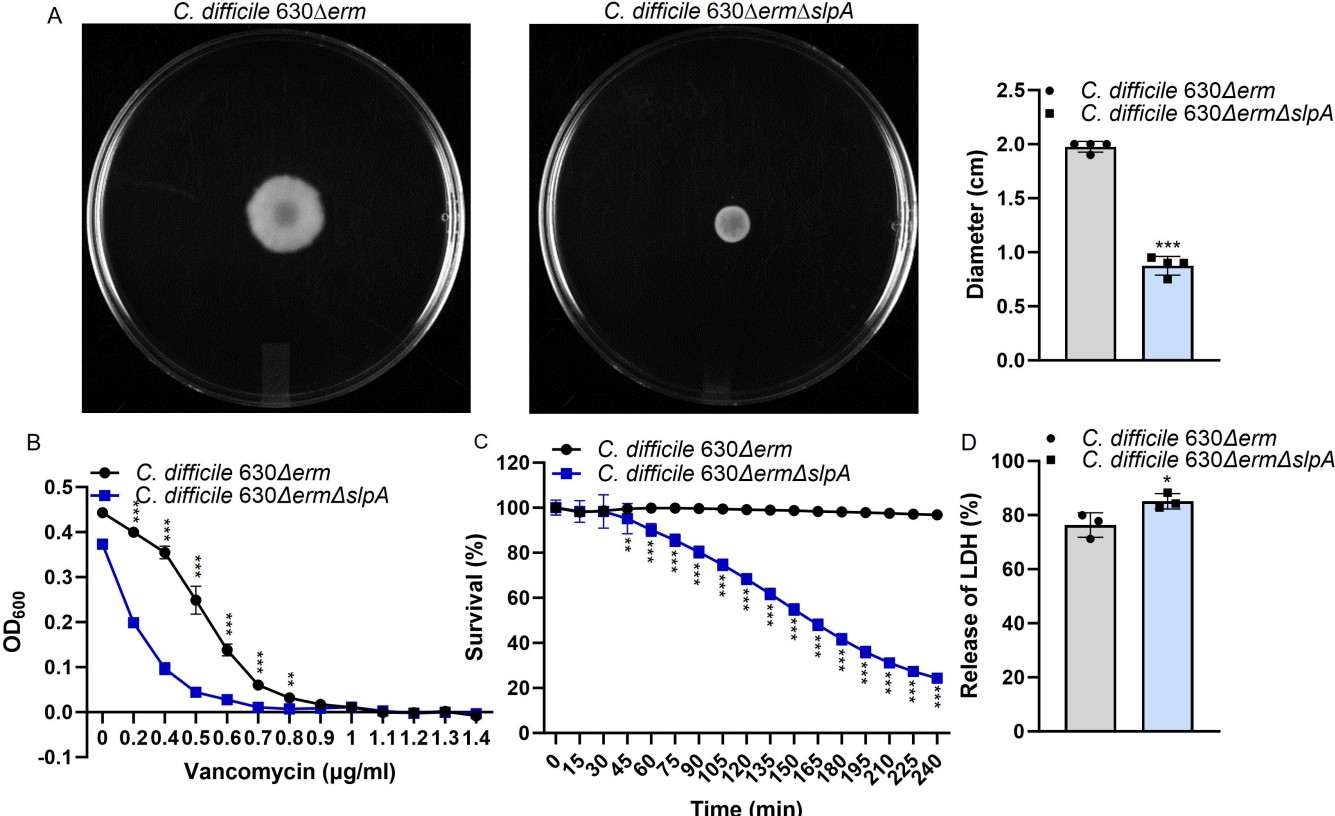

**FIG 3** Deletion of *slpA* led to motility inhibition with smaller swimming motility zone (*n* = 4) (A), increased sensitivity to vancomycin with less vancomycin MIC (*n* = 4) (B), prompted Triton X-100-induced autolysis (*n* = 4) (C) and more LDH release determined with LDH cytotoxicity assay (*n* = 3) (D) of *C. difficile* 630Δ*erm*Δ*slpA*. All these assays were repeated at least twice. *P*-values for differences between the wild-type and mutant strains, *$P < 0.05$, **$P < 0.01$, and ***$P < 0.001$.

## Roles of SlpA in adhesion of *C. difficile* to host cells

Adherence results of *C. difficile* mutant and the wild-type *C. difficile* strain to human epithelial cells (Caco-2) were demonstrated in Fig. 5. Deletion of *slpA* impaired adherence efficiency of *C. difficile* 630Δ*erm*Δ*slpA* to the host epithelial cell, achieving only 57.7% of that with the wild type, indicating the important role of SlpA in host-cell adherence of *C. difficile*.

## DISCUSSION

*C. difficile* infection is becoming a threat to public health, its cases are increasing, with poor prognosis, and becoming difficult to treat, especially during the coronavirus disease 2019 pandemic with increased use of antibiotics (22). Pathogenesis studies are urgently required to provide potential targets for CDI therapy. The major S-layer protein, SlpA, was supposed to play important roles in CDI pathogenesis, especially during host-pathogen interaction, and biofilm formation. However, the lack of isogenic *slpA* mutants has greatly hampered analysis of *C. difficile* S-layer functions (9). SlpA has always been thought to be essential until an insertion mutant (FM2.5) was obtained by chance (11). However, the insertion mutant could not remove *slpA* completely, and more details about roles of SlpA in *C. difficile* pathogenesis need to be uncovered. Therefore, it is necessary to generate *slpA* deletion mutants to decode roles of SlpA.

In this study, to challenge the possibility of removing the whole *slpA* gene, providing direct proofs supporting the roles of SlpA in CDI pathogenesis, we generated the first *slpA* deletion *C. difficile* mutant, *C. difficile* 630Δ*erm*Δ*slpA*, with CRISPR-Cas9 system (Fig. 1A). Complete removal of *slpA* was further confirmed with the loss of the two

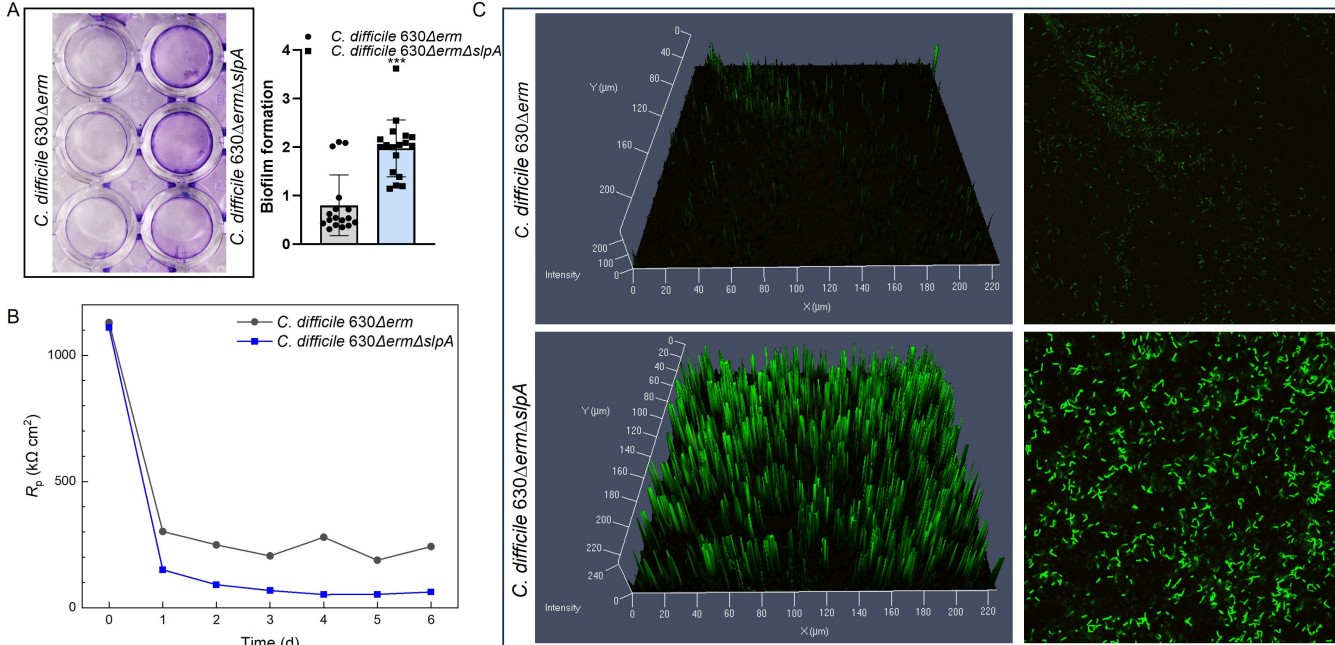

**FIG 4** Roles of SlpA in biofilm formation. (A) Deletion of *slpA* led to increased biofilm formation analyzed with crystal violet staining. (B) E-cells test monitored biofilm formation, and lower $R_p$, indicating more biofilm formation with the *slpA*-deleted mutant. (C) Bi-dimensional and three-dimensional biofilm structures were obtained using the LIVE/DEAD Biofilm Viability Kit and observed with CLSM. Strong fluorescence signal illustrated by *C. difficile 630ΔermΔslpA* further verified robust biofilm formation due to *slpA* deletion. Crystal violet staining assay and CLSM assay were repeated three times. E-cells test was performed once to support the other biofilm assays as a qualitative test. *P*-values for differences between the wild-type and mutant strains, ***$P < 0.001$.

major protein bands for HMW and LMW of SlpA (Fig. 1B). The successful generation of *slpA* deletion mutant further negated the previous knowledge which considered *slpA* essential. Similar with the first S-layer-null insertion mutant of *slpA* (FM2.5) (11) and the CRISPRi knockdown of *slpA* in R20291 (12), deletion of *slpA* resulted in more transparent colony morphology, inhibited growth (Fig. 2A), attenuated toxin production (Fig. 2C; Fig. S2B), and suppressed sporulation (Fig. 2D) of *C. difficile 630ΔermΔslpA*, verifying the important roles of SlpA in the life cycle of *C. difficile*. Moreover, major pathogenic factors such as interaction with host cells and biofilm formation of the *slpA* deletion mutant were also illustrated in this study and discussed below.

Interestingly, contrary to the weaker growth of *C. difficile 630ΔermΔslpA* determined with absorbance (Fig. 2A), SlpA deficiency increased cell density (CFU) of *C. difficile 630ΔermΔslpA* (Fig. 2B). This may be explained by the smaller bacteria and colony size of the mutant (Fig. S1). Moreover, the motility of the mutant was impaired (Fig. 3A). Flagella is primarily responsible for the swimming motility of *C. difficile* planktonic cultures (17); thus, the limited swimming motility of *C. difficile 630ΔermΔslpA* suggests potential expression differences of motility factors such as flagella and type IV pili due to deletion of *slpA*. The increased antibiotic sensitivity (Fig. 3B), Triton X-100-induced autolysis (Fig. 3C), and release of intercellular proteins (Fig. 3D) of *C. difficile 630ΔermΔslpA* also suggest changes of cell wall architecture which requires further study.

Another interesting morphology change is aggregation of *C. difficile 630ΔermΔslpA* in suspension (Fig. S2C). Overexpression of one of the cell wall proteins (CWPs), CwpV, was reported to prompt aggregation of *C. difficile*, changed colony morphology, and verified the N-terminal fragment was responsible for cell wall anchoring (23). There is an obviously overexpressed protein band (a little bit less than HMW of SlpA in lane 2 of Fig. 1B) from the mutant demonstrated the similar size of the N-terminal domain of CwpV (~42 kD) (23), indicating that the aggregation of mutant cells might be because of the high amount of CwpV. Further studies will be needed for identification of this band

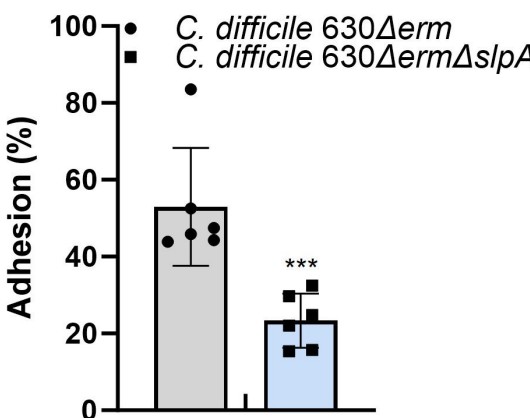

**FIG 5** Deletion of *slpA* resulted in decreased adhesion to Caco-2 cells. Adhesion tests were carried out in six replicates and repeated three times. *P*-values for differences between the wild-type and mutant strains, ***$P < 0.001$.

to verify the involvement of CwpV in *slpA* deletion-induced morphology changes. Other S-layer proteins around 15–25 kDa were also overexpressed in the mutant. Thus, it will be worthwhile to explore the functional interactions between SlpA and CwpV, other CWPs, as well as motility factors such as type IV pili and flagella in future studies. Moreover, identification of other CWPs, their expression differences due to SlpA deficiency, and their roles in *C. difficile* pathogenesis will provide new targets for CDI therapy.

As indicated before (17), along with decreased motility (Fig. 3A) and propensity to aggregate (Fig. S2C), a higher ability of biofilm formation (Fig. 4) was detected in the *slpA* deletion mutant. Increased biofilm formation was also observed with *cwp84* mutant, due to accumulation of uncleaved SlpA, or new role of Cwp84, which may be able to cleave other undetermined proteins (20). The robust biofilm of *C. difficile* 630Δ*erm*Δ*slpA* in this study excluded the possibility of uncleaved SlpA, but whether it was because of the decreased expression of Cwp84, new role of Cwp84, or danger signals due to deletion of *slpA* needs to be clarified further. eDNA was suggested to correlate with increase of biofilm formation (24), and the higher Triton X-100-induced autolysis rate demonstrated by *C. difficile* 630Δ*erm*Δ*slpA* indicated that more released eDNA may be another reason for the increase in biofilm formation of the mutant. Disruption of the *slpA* gene resulted in shorter, thin cells (Fig. S1B) and increased biofilm production (Fig. 4) compared to the wild type, indicating the stress of *C. difficile* 630Δ*erm*Δ*slpA* and emphasizing the effects of stress response on biofilm formation. The robust biofilm formation due to *slpA* deletion is a shock surprise because SlpA was indicated in prompting biofilm formation since the LMW cell wall protein SlpA was identified as the most abundant protein within the biofilm matrix unique bands (24), and incubation of *C. difficile* with anti-S-layer monoclonal antibodies significantly reduced biofilm formation (25). Therefore, the interesting opposite results in this study reinforced the necessity to find out more details about the roles of SlpA in biofilm formation and explore mechanisms in *C. difficile* biofilm formation.

SlpA was thought to be responsible for adhesion as, being expected, the decreased adhesion to Caco-2 cells verified the importance of SlpA during host-pathogen interaction. Considering the importance of SlpA in cell wall architecture (26), detection of *C. difficile* 630Δ*erm*Δ*slpA* cell surface structures will be helpful for revealing more precise targets responsible for pathogen-host interaction. The less toxin production due to SlpA deficiency further indicated the importance of SlpA in CDI pathogenesis. Future *in vitro* and *in vivo* studies with the wild type, *C. difficile* 630Δ*erm*Δ*slpA*, a complemental *C. difficile* 630Δ*erm*Δ*slpA* expressing *slpA*, as well as *C. difficile* overexpressing *slpA* will be included for discovering additional mechanisms under effects of SlpA on CDI pathogenesis.

In conclusion, we generated the first *slpA* deletion *C. difficile* strain in this study, further providing solid proof against the essential role of *slpA* in *C. difficile*. We

investigated effects of SlpA on morphology, key life cycle steps of *C. difficile*, antibiotic sensitivity, Triton X-100-induced autolysis, release of intracellular protein, biofilm formation, and adhesion to host cells. These results will provide direct proofs for previously indicated and new roles of SlpA in *C. difficile* pathogenesis, which will facilitate our future investigations for new targets for developing vaccines, efficient therapeutic agents, and novel intervention strategies to combat CDI.

## MATERIALS AND METHODS

### Bacterial strains and growth conditions

Strains used in this study were listed in Table 1. *Escherichia coli* strains were grown in lysogeny broth (LB) medium at 37°C. Where appropriate, kanamycin (Kan, 50 µg/mL) and chloramphenicol (Cm, 20 µg/mL) would be supplemented into LB medium for plasmids selection. *C. difficile* strains were cultivated in an anaerobic chamber (90% $N_2$, 5% $CO_2$, 5% $H_2$ by volume) at 37°C in Brain Heart Infusion (BHIS, supplemented with 5 g/L of yeast extract and 1 g/L of L-cysteine) (27). Where indicated, thiamphenicol (Tm, 15 µg/mL), D-cycloserine (250 µg/mL), cefoxitin (8 µg/mL), anhydrotetracycline (aTet, 100 ng/mL), and taurocholate (0.1% wt/vol) were supplemented to BHIS medium as needed.

### Generation of *slpA* deletion mutant

#### *Plasmid construction*

Primers and gblock sequences used for plasmids construction were listed in Table 2. Gblock containing gRNA sequence and 20-nt guiding sequence targeting on the *slpA* gene (CD630_27930) was inserted to the *Kpn*I and *Mlu*I of pJK02 (Addgene plasmid #105133), generating the plasmid pSlpA1 and confirmed by colony PCR (cPCR) with primers SW15 and SW17 according to previous protocols (30). The homology arm sequences (~500 bp each) around the *slpA* open reading frame (ORF) were amplified with primer pairs SW1 and SW2, SW3, and SW4. The final plasmid pSlpA2 was obtained through the insertion of the homology arm sequences (fused together with primers SW1 and SW4) into the *Not*I and *Xho*I site of pSlpA1.

#### *Conjugation*

Transformation of plasmid into *C. difficile* was carried out by conjugation (27, 31). pSlpA2 was transformed into the donor strain *E. coli* CA434 (HB101 carrying R702) generating the recombinant *E. coli* CA434 strain which was cultivated overnight in LB medium supplemented with Kan (50 µg/mL) and Cm (25 µg/mL). One milliliter of cell culture was harvested and washed once with 1 mL of BHIS through centrifugation at 5,000 *g* for 2 min. Meanwhile, 500 µL overnight culture of *C. difficile* 630Δ*erm* was harvested and heat

**TABLE 1** Strains and plasmids used in this study

| Name | Characteristics | Sources or references |
|---|---|---|
| Strain | | |
| *E. coli* | | |
| DH5α | F⁻ *endA1 glnV44 thi-1 recA1 relA1 gyrA96 deoR nupG purB20* φ80d*lacZ*ΔM15 Δ(*lacZYA-argF*) U169 hsdR17($r_K^- m_K^+$) λ⁻ | NEB |
| CA434 | thi-1 hsdS20 (r-B, m-B) *supE44 recAB ara-14 leuB5proA2 lacY1 galK rpsL20* (str^R) *xyl-5 mtl-1 plasmid R702* (Tra⁺ Mob⁺ IncP Km^R Tc^R Sm^R Su^R Hg^R) | (28) |
| *C. difficile* | | |
| 630Δ*erm* | Clinical isolate with deletion of the erythromycin marker | (29) |
| 630Δ*erm*Δ*slpA* | 630Δ*erm*Δ*slpA* | This work |
| Plasmid | | |
| pJK02 | pCD6 ori Cm^r PTetR::Cas9 gRNA; *TraJ* | (30) |
| pSlpA1 | pJK02::20nt-gRNA | This work |
| pSlpA2 | PSlpA1::homology arm sequences | This work |

**TABLE 2** Primers and gblock sequences used in this study

| Primer | Sequence (5´–3´) |
| --- | --- |
| SW1 | GGAAACAGCTATGACCGCGGCCGCCATCTCTGCTATCTTTCCTTG |
| SW2 | AATGTTGGGAGGAATTTAAGGGCTTCTCTCATGAGAAGTC |
| SW3 | GACTTCTCATGAGAGAAGCCCTTAAATTCCTCCCAACATT |
| SW4 | TCGCGCATGTCTGCAGGCCTCGAGAGTATATGGCTGATAAAGGTG |
| SW9 | CAACTAGAGTTTTACCTTCAC |
| SW10 | GAAAATTTACAAGGAATGGC |
| SW13 | GACCACTACTTTGCAAGTGT |
| SW14 | GTCACATACTGCGTGATGAA |
| SW15 | CCACTTACAGCTACAGCACT |
| SW17 | GTATCTGCGCTCTGCTGAAG |
| Gblock SW1 | gtgtgctataattaaactgtaaaggtaccccacttacagctacagcactgtttttagagctagaaatagcaagt-taaaataaggctagtccgttatcaacttgaaaaagtggcaccgagtcggtgctttttttctatggagaaatcta-gatcagcatgatgtctgactagacgcgtaagctctgcaactatttttagatgg |

treated at 52°C for 5 min. After cooling, the *C. difficile* 630∆*erm* strain was added to the pellet of the donor recombinant strain, mixed well, and dropped onto pre-reduced BHIS agar plates by dropping 10 20 µL drops of the mixture. After 8-h cultivation, growth was harvested with 1 mL of BHIS medium, and 100 µL of the resuspended growth was plated onto BHIS selective plates containing D-cycloserine (250 µg/mL), cefoxitin (8 µg/mL), and Tm (15 µg/mL), and incubated anaerobically at 37°C for 24–72 h.

## Induction of the CRISPR-Cas9 system and mutant screening

Single colonies from the BHIS screening plates were picked for cPCR using primers SW13 and SW14 to confirm the presence of the plasmid. The recombinant colony was cultivated in BHIS medium supplemented with Tm (15 µg/mL) to permit efficient homologous recombinant, and the overnight culture was inoculated (5% inoculum) into fresh BHIS medium with Tm and aTet (100 ng/mL) for 6 h for Cas9 induction, followed by being spread onto BHIS plates supplemented with Tm and aTet. Mutants were screened through cPCR using the primer pair (SW9 and SW10) annealing to the chromosomal loci beyond the homologous recombination regions.

## Plasmid curing

Colonies demonstrating only the mutant band were picked and cultivated in BHIS medium without antibiotics. Subcultures were carried out every 8 h for 10 cycles, followed by being spread onto BHIS to obtain single colonies, which were then replica plated onto BHIS agar and BHIS agar with Tm plates. Colonies that grew well on the BHIS agar but failed to grow in the presence of antibiotics were considered as putative cured derivatives. Plasmid loss in the cured derivatives was confirmed by cPCR using primers SW13 and SW14. cPCR with primers SW9 and SW10 was conducted to further verify the existence of the desirable mutation, and the mutant was named as *C. difficile* 630∆*erm*∆*slpA*.

## Extraction of S-layer proteins and SDS-PAGE

S-layer protein fractions were extracted using low-pH glycine as described by Amirkamali et al. (32) and Calabi et al. (33) with minor modifications. Briefly, logarithmic (log) phase *C. difficile* strains were collected and centrifuged (3,000 *g*, 20 min), and pelleted cells were resuspended in 0.2 M glycine (pH 2.2, Sigma-Aldrich) and kept at room temperature with shaking for 30 min. The bacterial cellular debris was removed by centrifugation (16,000 *g*, 15 min, 4°C), and the supernatants containing SlpA were collected, neutralized with 2 M Tris, and evaluated on SDS-PAGE gels followed by Coomassie blue staining.

## Determination key life cycle steps of *C. difficile*

### *Growth*

*C. difficile* 630Δ*erm*Δ*slpA* and the wild type *C. difficile* 630Δ*erm* strains were inoculated in BHIS medium, and overnight cultures were transferred (2% inoculum) into 96-well plates with fresh BHIS medium. Growth (changes of OD600) of *C. difficile* strains were monitored at 37°C in a Synergy H1 microplate reader (BioTech). CFU of *C. difficile* strains were compared by taking samples at 12 h, 24 h, 36 h, and 48 h, followed by being cultivated on BHIS agar plates. Images of BHIS plates with colonies cultivated after 12 h were taken using a ChemiDoc Gel Documentation System (Bio-Rad). Colonies at logarithmic growth phase were also checked under microscope (Nikon Eclipse E600), and length of bacteria was measured with image J software.

### *Toxin production*

Production of *C. difficile* toxins (TcdA/B) in wild-type and mutant strains were measured with *C. difficile* Toxin A & B ELISA Kit from EDI (Epitope Diagnostics). The wild-type and *slpA* deletion *C. difficile* strains were cultivated in BHIS medium. Samples were taken at 24 h and 48 h. After centrifugation (13,000 *g*, 5 min, 4°C), the supernatants were collected for ELISA as external toxin production. The pellets were resuspended in phosphate buffered saline (PBS), sonicated, and centrifuged to determine intracellular toxins.

### *Sporulation*

*C. difficile* strains (the wild-type and *slpA*-deleted mutant) were cultivated in Clospore media for 5 d (34, 35) for spore forming. Spores were collected, washed with water, and heat treated at 65°C for 30 min to kill the remaining vegetative bacilli, and enumerated by cultivation on taurocholate, cefoxitin, cycloserine, and fructose agar plates. Sporulation was evaluated as the ratio of heat-resistant spores per total viable count.

## MIC determination

Sensitivity to vancomycin was determined with serial dilution and gradient vancomycin (36). The wild-type *C. difficile* 630Δ*erm* and *C. difficile* 630Δ*erm*Δ*slpA* strains were cultivated in a 96-well plate with BHIS liquid medium supplemented with vancomycin (0–1.4 µg/mL). Growth (change of OD600) of *C. difficile* strains was monitored at 37°C in a Synergy H1 microplate reader (BioTech), and growth after 10-h cultivation was used to compare their sensitivity to vancomycin.

## Motility assays

Motility behavior of *C. difficile* was studied by performing swimming motility assays (37). Cultures of *C. difficile* were grown to mid-exponential phase for 8 h in BHI broth under anaerobic condition at 37°C. Swimming soft agar (BHI broth medium containing 0.3% (wt/vol) agar) plates were stab-inoculated with 3 µL of a mid-exponential growth culture of each strain at the same cell density. After 48-h incubation at 37°C, swimming motility was quantitatively determined by measuring the radius.

## Biofilm formation assays

### *CV staining assay*

The wild-type *C. difficile* 630Δ*erm* and the mutant *C. difficile* 630Δ*erm*Δ*slpA* strains were cultivated in BHISG (BHIS supplemented with 0.1 M glucose) in tissue culture-treated 96-well polystyrene plates (38) and cultivated at 37°C anaerobically. After 24-h cultivation, OD600 was determined with a Synergy H1 microplate reader (BioTech). Then, the planktonic phase was removed, and the biofilm was washed and stained with CV, and biomass was measured at OD595 (38, 39). The ability of biofilm formation was compared using the ratio of absorbance after to before CV staining.

## E-cells assay

Many biofilms exhibited corrosivity against various metals such as carbon steel and stainless steel, albeit very low in many cases to have practical importance in corrosion damages (21). However, extremely sensitive electrochemical measurements can be used to correlate transient corrosion rate with biofilm growth and antimicrobial treatment efficacy in real time. To monitor *C. difficile* biofilm growth and health, the newly developed biofilm test kit based on a miniature electrochemical cell using a 304 stainless steel (1 cm$^2$ exposed working surface) as working electrode and graphite counter electrode/pseudo reference electrode in a standard 10 mL serum vial was used. Each vial with the two solid-state electrodes was autoclaved, sealed in a N$_2$ chamber, and injected with 7 mL corresponding culture medium pre-inoculated with wild-type or *slpA*-deleted *C. difficile* mutant (10$^6$ cfu/mL). The biofilm test kit vials were incubated at 37°C and scanned every 3 h using an electrochemical workstation (model CS300, Corrtest) to monitor decrease of corrosion resistance (R$_p$) using linear polarization resistance (40). The initial R$_p$ decrease (corrosion rate increase) was indicative of the formation of a corrosive biofilm on the 304 stainless steel working electrodes. When R$_p$ starts to reach the minimum level, it means the biofilm reaches maturity.

## CLSM assay

Three-dimensional structure of biofilm and dead/live cells in biofilm was analyzed using CLSM at the Microscope Facility of Ohio University (41, 42). Biofilms grew on sterile glass bottom dishes for 24 h, and the mature biofilms were washed using a MgSO$_4$ solution, stained with acridine orange, and detected under CLSM. Dead/live cells were observed after staining with LIVE/DEAD BacLight Bacterial Viability Kit L7012 (Life Technologies) for green live cells at an excitation wavelength of 488 nm and red dead cells at 559 nm.

## Triton X-100-induced autolysis and LDH secretion

For Triton X-100-induced autolysis (43), *C. difficile* strains (OD600 ~1) were resuspended in 50 mM potassium phosphate buffer (pH 7.0) containing 0.01% Triton X-100 and incubated at 37°C. The OD600 was determined every 5 min for 4 h in a Synergy H1 microplate reader (BioTech). For release of LDH, cells were cultivated in BHIS for 24 h and centrifuged, and then the pellets were resuspended in PBS and sonicated. Extracellular LDH (from supernatant) and intercellular (from pellet) LDH were determined with the CyQUANT LDH Cytotoxicity Assay Invitrogen/thermos (C20300) (Thermo Fisher) according to the instructions of the manufacturer (44). Release of LDH from different strains was compared based on the percentage of extracellular LDH to total LDH (extracellular plus intracellular).

## Adherence assays

To demonstrate the important role of SlpA in host-cell adherence, attachment of *C. difficile* 630Δ*erm*Δ*slpA* and the wild type to human epithelial cells was investigated as described by Merrigan et al. (6). The human colon carcinoma cell line Caco-2 (ATCC) was used between passages 45 and 50. Cells were cultured in Eagle's minimum essential medium (EMEM) with Earle's Balanced Salt Solution (EBSS) supplemented with with 2 mM glutamine, 1% non-essential amino acids (NEAA), and 10% fetal bovine serum as recommended. Moreover, 100 IU/mL of penicillin and 100 mg/mL streptomycin were also added into the medium to inhibit growth of bacteria. After 3-d incubation at 37°C in a 5% atmosphere, cells grown as confluent monolayers (approximately 2.25 × 10$^6$ cells) were cultured in antibiotic and serum-free medium for about 24 h, and then were introduced into the anaerobic chamber for adherence assays. Meanwhile, *C. difficile* strains at the exponential phase were harvested, washed once, and resuspended in anaerobic EMEM-Ca (25 mM). Subsequently, the medium was removed from six-well plates, and *C. difficile* strains were added at a multiplicity of infection of 20 in a total volume of 2 mL anaerobic EMEM-Ca (25 mM). After a 40-min incubation, host cells and

adherent bacteria were washed twice with 1 mL of reduced PBS, scraped, vortexed, serially diluted, and spread onto BHIS agar plates. The adherent *C. difficile* strains were enumerated after 24- to 48-h incubation, and the adherence efficiency was calculated as the ratio of recovered *C. difficile* strains to the total input strains.

## Statistical analysis

Statistical differences among groups were analyzed using two-tailed unpaired Student's *t*-test and/or analysis of variance. All the quantitative assays were performed at least two to three times with at least three replicates each time, and the values presented in graphs are means ± standard error of means or means ± SEM. GraphPad (Prism9) was used for making figures. $P < 0.05$ was considered statistically significant.

## ACKNOWLEDGMENTS

We thank Dr. Shonna McBride from Emory University for sharing the *C. difficile* 630Δerm strain with us.

We thank the Start-up funding and Research and Scholarly Awards Committee (RSAC) pilot grant support from Ohio University Heritage College of Osteopathic Medicine.

## AUTHOR AFFILIATIONS

[1]Department of Biomedical Sciences, Ohio University Heritage College of Osteopathic Medicine, Ohio University, Athens, Ohio, USA
[2]Infectious and Tropical Disease Institute, Ohio University, Athens, Ohio, USA
[3]Department of Chemical and Biomolecular Engineering, Institute for Corrosion and Multiphase Technology, Ohio University, Athens, Ohio, USA

## AUTHOR ORCIDs

Shaohua Wang (i) http://orcid.org/0000-0002-8548-1602

## FUNDING

| Funder | Grant(s) | Author(s) |
| --- | --- | --- |
| Ohio University (OU) | START UP, RSAC | Shaohua Wang |

## AUTHOR CONTRIBUTIONS

Shaohua Wang, Conceptualization, Funding acquisition, Investigation, Methodology, Project administration, Resources, Supervision, Writing – original draft | Maria C. Courreges, Data curation, Methodology | Lingjun Xu, Formal analysis, Methodology | Bijay Gurung, Methodology | Mark Berryman, Methodology | Tingyue Gu, Supervision, Writing – review and editing

## ADDITIONAL FILES

The following material is available online.

## Supplemental Material

**Supplemental figures (Spectrum04005-23-s0001.docx).** Fig. S1 and S2.

## Open Peer Review

**PEER REVIEW HISTORY (review-history.pdf).** An accounting of the reviewer comments and feedback.

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
