## [Reviewer comments · Microbiology Spectrum]

Microbiology Spectrum

Revealing roles of S-layer protein (SlpA) in *Clostridioides difficile* pathogenicity by generating the first *slpA* gene deletion mutant

Shaohua Wang, Maria Courreges, Lingjun Xu, Bijay Gurung, Mark Berryman, and Tingyue Gu

Corresponding Author(s): Shaohua Wang, Ohio University Heritage College of Osteopathic Medicine

Review Timeline:

Submission Date:	November 21, 2023
Editorial Decision:	February 13, 2024
Revision Received:	March 8, 2024
Accepted:	April 16, 2024

Editor: Meera Unnikrishnan

Reviewer(s): The reviewers have opted to remain anonymous.

Transaction Report:

DOI: <https://doi.org/10.1128/spectrum.04005-23>

Re: Spectrum04005-23 (Revealing roles of S-layer protein (SlpA) in *Clostridioides difficile* pathogenicity by generating the first *slpA* gene knockout mutant)

Dear Dr. Shaohua Wang:

Thank you for submitting your manuscript to Spectrum.

My apologies for the delay in getting this decision back to you.

We have received comments back from two expert reviewers which are included below. Although they find the experimental data in the paper robust, overall a number of points have been raised with regard to experimental description, interpretation and discussion of results. Additionally, further clarification is required for some of the data presented. Please submit a revised manuscript addressing the Reviewers' concerns.

As suggested by the reviewers, for this manuscript to be considered for publication, an extensive revision of the text for language would be required. I recommend that you ask a colleague of yours who is a native English speaker to read and provide you some feedback on the writing. You are also welcome to use one of the services here: <https://journals.asm.org/content/language-editing-services> to Microbiology Spectrum.

Revision Guidelines

Thank you for submitting to Spectrum.

Sincerely,
Meera Unnikrishnan
Editor

Reviewer #1 (Comments for the Author):

This manuscript by Wang et al describes the construction and characterisation of a 630erm slpA deletion strain. Over the years many groups have tried, without success, to construct such a strain, so this will be of great interest to the community. The data presented appears solid and interpretation is not over the top. However, I do have some concerns about how the authors have placed their work into the wider context of the field and, in particular, the downplaying of previous work on slpA in *C. difficile*. Although this is the first slpA deletion that I am aware of, it is not the first S-layer null mutant and many of the phenotypes described here have been described previously. Despite this, the similarities to previously published work are not highlighted by the authors.

Kirk et al 2017 described colony morphology, growth, sporulation and toxin expression differences essentially identical to those described here but that is not acknowledged at all. Müh et al also observed the same colony morphology differences using CRISPRi knock down, confirmed the sporulation effects and also noted a propensity to cell lysis.

Line 27: "was constructed" - as written this implies that you've built the CRISPR-Cas9 system rather than using a previously published vector. Please rephrase.

Line 28: "knocked out" - I would suggest changing this to deleted here and elsewhere. The novelty here is the true deletion of slpA.

Line 65: Reference 11 here is not the original source. The slpA frameshift mutant was first described in Kirk et al 2017.

Line 66: "while the frameshift was suppressed during infection" I think this is referring to the mutant being avirulent but it's really hard to make sense of this as written. Please rephrase.

Lines 68-69: Müh et al also described sporulation defects and propensity to lysis similar to what you describe in this manuscript.

Line 103: the indicated band is definitely not Cwp84, at least not solely. Cwp2, Cwp66 and Cwp84 co-migrate and are very difficult to separate on SDS PAGE. Of the three, Cwp84 is by far the least abundant.

Line 120: there is a difference in response to vancomycin here but the significance is hard to justify - a 0.2 ug/ml difference in MIC hardly seems worth reporting.

Line 122: I'm not sure why you are referring to this process as autolysis. Autolysis is an active process mediated by cell wall hydrolases and that does not appear to be what you are seeing here.

Lines 127-128: aggregation is not synonymous with biofilm formation as implied here.

Lines 136-138: I may be misunderstanding what's presented here but fluorescence intensity is not a good read out of biofilm thickness. As you have performed confocal microscopy you can presumably see the thickness in Z stacks.

Line 143: "direct proof for the key role of SlpA in host-cell adherence" - this is a significant over-interpretation. The slpA mutant is lacking the dominant surface protein, has a surface with completely different biochemical characteristics and has likely changed expression and localisation of multiple other cell surface proteins, several of which have been implicated in adherence. Ascribing a direct role for SlpA in adherence is not justified by these data.

Line 152: 11 is the wrong reference again.

Line 153: "has not been characterised [sic] well" - this is simply not true. The previously described S-layer null mutant that you are referring to here has been extensively characterised in at least two papers (Kirk et al 2017 and Ormsby et al 2023), revealing many of the same phenotypes that you have described here, in addition to several others that you have not tested.

Lines 159-160: No, Kirk et al 2017 already demonstrated that slpA is not essential and this was subsequently supported by CRISPRi by Müh et al in 2019.

Line 161: "growth of which was not obviously affected" - not correct. Kirk et al noted that the slpA frameshift mutant reached a maximum stationary phase OD of 2.2 compared with 3.2 for the wild type strain. This is essentially identical to the 60% defect you describe here.

Line 178-180: If the 42 kDa band were indeed the C terminal fragment of CwpV then a similarly intense band for the larger N terminal fragment would also be apparent.

Line 197: the difference is length fairly small and I would not refer to these as filamentous.

Line 202-203: the reference to the *D. radiodurans* SlpA here is a little odd as the proteins have absolutely no similarity.

Line 213: again has not been considered essential for many years.

Line 238: References 27 and 28 are not the appropriate original sources for this method.

Lines 257-258: "subcultured in... 10 continuously cycles" - I don't follow what you've done here. Can you please rephrase?

Line 266 onwards: Amirkamali is not the original source for this method.

Line 290: 65C for how long?

Line 294: How were these inoculated? Very important as that can have a dramatic effect on MIC. "Growth... monitored" - only endpoints are shown so presumably all of these were in stationary phase?

Line 337: again I would argue that this is not autolysis.

Figures and legends

Number and type of replicates should be added to all legends.

Figure 1: As noted above this is likely not Cwp84.

Figure 2: Very little detail in the legend. More here would save the reader referring back and forth to the methods and would make the figure much more understandable.

Figure 3: very little detail here. What does the dotted line in B represent? Also the measurement here is presumably max growth expressed as a percentage of the control. Survival is not correct when talking about MIC.

Figure 4: B has no error bars - single measurements? On rereading C I can't figure out if this is actually representing intensity across a Z stack or not. The legend needs a lot more information to aid interpretation.

Figure S1: B is cell length not size presumably. Are these images phase contrast? If so they are extremely over-saturated.

Figure S2: panel A would be much better merged with Fig. 2B. What is shown in S2B? The legend only says toxins [sic] production at 24 h and 48 h cultivation - not nearly enough detail incidentally. Fig 2C apparently also shows "less toxins production" at 24 and 48 h. So it seems these are showing the same thing, yet the values are completely different - there's > an order of magnitude difference in the wt at 48 h.

Merging Figure 2B and S2A would help reader understanding.

Minor typographical issues

Line 35: hospital acquired infection

Line 45: spore forming

Line 54: SLPs are derived

Line 101: loss of

Line 110: slpA not SlpA

Line 114: toxin production

Line 175, 183 and 185: CWPs

Line 189: cwp84 italics

Line 221: coli

Line 226: anhydrotetracycline

Lines 308-309: in tissue culture

Line 311: biofilm was washed

Reviewer #2 (Comments for the Author):

The data presented in this paper is compelling, supporting and extending previous observations regarding the contribution of the SlpA on a wide variety of phenotypes including toxin production and the capacity of the strain to sporulate.

However, to improve the paper, there is a need to contextualise the data presented - how can it be linked to the loss of the S layer and what additional data would need to be included to support these observations. Description of the results is currently very limited and needs to be extended to include a greater explanation of ALL the results presented and what they mean. This could be supported in the discussion with a further linkage to existing literature. If the aim of the paper is to highlight the potential and effective use of CrispR Cas as a mutational system, then generation of an additional mutant in a strain with a different SlpA type is warranted.

A review of the written English is required throughout as there are numerous spelling mistakes and a number of poorly written passages.

Review: Revealing roles of S-layer protein (SlpA) in *Clostridioides difficile* pathogenicity by generating the first slpA gene knockout mutant.

This paper, by Shaohua Wang et. al., offers an insight into the role of SlpA in *C. difficile* pathogenesis through the deletion, via CrispR/Cas mutagenesis, of the entire *slpA* of 630 Δ erm. This genetic modification resulted in the demonstrable loss of expression of both the post-translational cleaved HMW and LMW proteins in low pH glycine extracts from this bacterium. The work then focuses on characterisation of differences between the parental and modified strain, in the context of several relevant features of pathogenesis including growth rates, toxin production, sporulation and biofilm formation.

Main Comments

The authors highlight that the generation of this mutant using CrispR/cas nullifies the argument that SlpA is essential for survival, a concept that was promoted due to the failure of other systems to generate a site-specific mutant. However, this should be qualified as while no site-specific mutant has been produced, a naturally occurring mutant has been isolated. Regarding its essentiality *in vivo*, this remains unclear from this manuscript as no testing was performed. Worryingly, data in Figure S2, suggests that there is a significant loss of mutant viability after 48h, which would indicate that the SlpA expression have relevance for even long-term survival *in vitro*.

While the authors report their direct observations regarding the behaviour of the mutants *in vitro*, the explanation and interpretation of the phenotypes observed is limited, especially in the context of the presence and absence of an S layer. It is reported that the mutant sporulates poorly and shows limited toxin production, however, there is no wider attempt to explain why this might be the case. Given that the mutant appears to show limited viability after 48h does this reflect the fact that a high proportion of the bacteria are no longer viable or they cannot sporulate effectively? Interestingly, several of the features described here – including reduction in colony size, toxin production, sporulation defects and increased sensitivity to lysis have been described elsewhere in the literature in the context of a naturally occurring SlpA mutant, R20291 FM2.5. In the reviewed manuscript, this mutant is dismissively described as poorly characterised, however, given the similarities in several features, acknowledging this work would strengthen the manuscript; given the additional data provided that may help to explain differences in toxin production and sporulation described here.

Other data is simply reported without contextualization. What is the relevance of loss of motility – Is it associated with loss of membrane stability – are the flagellar still made – can they be recovered from the supernatant of growth medium. There should be an expansion of either the description of these results, which are currently very limited, or additional speculation added to the discussion to provide greater overview of the observations. What is the relevance of enhanced susceptibility to killing by vancomycin and Triton X ?

In parallel, there is no real explanation as to why the OD values and viability of the mutant do not correlate. If the bacterial cells are smaller in size as indicated (and which data in Fig S1 supports), the higher density of bacteria as indicated by the viability counts would still be expected to increase light deflection by the culture. Statistically, is the difference in viability

significant? The differences in Figure 2B are enhanced by the choice of a linear rather than logarithmic scale for this data. However, if true, the difference in OD/viability raises questions regarding the standardisation of inoculum for the assays described. For example, for the adhesion assay, the paper indicates mid exponential cells were added at a ratio of 20:1, however, it was not clear how this was standardised to provide the same starting inoculum. Similarly given the observed loss of viability of the mutant over time (48h) in vitro (Fig S2), could the reported differences in adhesion between the mutant and parental strain be linked to loss of viability and not differences in capacity to adhere? Was this considered?

For biofilm production, it would be useful to include electron micrographs of the biofilm structures generated by the mutant and parental strains. While data is very supportive of this phenotype, increased crystal violet staining could reflect enhanced active biofilm formation by the mutant, or modifications to the surface charge of the bacterial surface, resulting in aggregation rather than biofilm formation. While other methods for biofilm measurement are included indicated either these are no standard for biofilm measurement (F_p) or often suffer from the capacity to penetrate the biofilm structure effectively (live/dead stain).

Lastly, given that the highlight of this paper is the generation of a site directed SlpA mutant using Crisp/Cas, an attempt, or at least tried attempt to generate at least a second mutant from a strain expressing a different SlpA type would enhance the paper. The feasibility of this may be dependent on existing antibiotic resistances in the strains, but the generation and testing of an additional mutant would provide a broader evaluation of the technology and greater understanding of the essentiality of this gene. In particular, the ease by which similar mutations in other SlpA types could be generated could provide important information.

Minor points.

The language used in the introduction, results and discussion is naïve and ambiguous in places. In contrast the methods are well written, although additional information on standardisation of the inoculum in various assays would be useful. There are multiple spelling and grammatical errors throughout the manuscript that require modification.

In the introduction, there is a strong statement indicating that adhesion is critical for *C. difficile* pathogenesis and yet this has not definitively been shown at least *in vivo*.

In a number of places, there are references to intercellular proteins. It is not clear what location this relates

One main

Responses to reviewers:

We appreciate the reviewers' time and dedication to giving feedback and are grateful for insightful suggestions, which have helped us improve our manuscript substantially. This revised version of the manuscript addresses all the comments provided by the reviewers. The changes made are highlighted in yellow within the manuscript. Please check below for the responses to each comment from the reviewers.

Reviewer #1 (Comments for the Author):

This manuscript by Wang et al describes the construction and characterisation of a 630erm *slpA* deletion strain. Over the years many groups have tried, without success, to construct such a strain, so this will be of great interest to the community. The data presented appears solid and interpretation is not over the top.

Thanks for recognizing the novelty of our study, and your positive comments on our results.

However, I do have some concerns about how the authors have placed their work into the wider context of the field and, in particular, the downplaying of previous work on *slpA* in *C. difficile*. Although this is the first *slpA* deletion that I am aware of, it is not the first S-layer null mutant and many of the phenotypes described here have been described previously. Despite this, the similarities to previously published work are not highlighted by the authors.

We totally agree with the reviewer's concerns. We did include the previous studies in generating S-layer null mutant and *slpA* knockdown mutants. As suggested, in the revised version, we added more details highlighting and comparing phenotypes between those mutants.

Kirk et al 2017 described colony morphology, growth, sporulation and toxin expression differences essentially identical to those described here but that is not acknowledged at all. Müh et al also observed the same colony morphology differences using CRISPRi knock down, confirmed the sporulation effects and also noted a propensity to cell lysis.

Thanks! We discussed similar findings in line 160-167.

Line 27: "was constructed" - as written this implies that you've built the CRISPR-Cas9 system rather than using a previously published vector. Please rephrase.

Thanks for your careful check, we changed the description to "In this study, the whole *slpA* gene was successfully deleted for the first time via CRISPR-Cas9 system." in line 27.

Line 28: "knocked out" - I would suggest changing this to deleted here and elsewhere. The novelty here is the true deletion of *slpA*.

Good suggestion, thanks! We have replaced "knocked out" with "deleted" in the manuscript.

Line 65: Reference 11 here is not the original source. The *slpA* frameshift mutant was first described in Kirk et al 2017.

Thanks, we have cited the direct reference from Kirk et al. 2017. in line 64.

Line 66: "while the frameshift was suppressed during infection" I think this is referring to the mutant being avirulent but it's really hard to make sense of this as written. Please rephrase.

Thanks for your good suggestions, we modified the sentence to "and the the framshift mutant is avirulent during infection".

Lines 68-69: Müh et al also described sporulation defects and propensity to lysis similar to what you describe in this manuscript.

We added these results in line 68-69 as "Based on this CRISPRi system, the expression of SlpA was knocked down, and the *slpA*-depleted cells demonstrated reduced sporulation and increased lysozyme sensitivity (12)."

Line 103: the indicated band is definitely not Cwp84, at least not solely. Cwp2, Cwp66 and Cwp84 co-migrate and are very difficult to separate on SDS PAGE. Of the three, Cwp84 is by far the least abundant.

Appreciate your concerns. As we are not able to distinguish this band based on the size, we removed the indication for CWP84 decrease, and changed the description as "Along with disappearance of SlpA bands, other cell wall proteins including those around 50-75 kDa were lower expressed, and those around 15-50 kDa were higher expressed". Moreover, we modified the Fig. 1 and the legend accordingly.

Line 120: there is a difference in response to vancomycin here but the significance is hard to justify - a 0.2 ug/ml difference in MIC hardly seems worth reporting.

We agree that the 0.2 ug/ml difference in MIC is slight. As the sensitivity to different vancomycin concentrations are still obvious, we modified the description to "*C. difficile* 630 Δ *erm* Δ *slpA* exhibited a vancomycin MIC of 0.8 μ g/ml, slightly (0.2 μ g/ml) less than that for *C. difficile* 630 Δ *erm*, but *C. difficile* 630 Δ *erm* Δ *slpA* was more susceptible to vancomycin at sub-MICs (Fig. 3B)." in line 119-121.

Line 122: I'm not sure why you are referring to this process as autolysis. Autolysis is an active process mediated by cell wall hydrolases and that does not appear to be what you are seeing here.

Thanks for your great question! Different from the spontaneous autolysis, we used Triton X-100 to stimulate autolysis. To be clear, we modified all the autolysis to "Triton X-100-induced autolysis".

Lines 127-128: aggregation is not synonymous with biofilm formation as implied here.

Thanks for your rigorous advice, we modified it as a potential indication, and added a citation to support this indication (line 129-130).

Lines 136-138: I may be misunderstanding what's presented here but fluorescence intensity is not a good read out of biofilm thickness. As you have performed confocal microscopy you can presumably see the thickness in Z stacks.

Thanks for your thoughts. CLSM images are used to support biofilm sessile cell information in both 3-D and 2-D modes. In 3-D mode, the biofilms appear to be uneven. Thus, z-axis height

ratio is not a good quantitative measurement. Although fluorescence intensity is not very accurate, it is still widely used quantitatively by many researchers. To decrease the confusion, we removed “thickness” from the sentence, and focused on fluorescence intensity.

Line 143: "direct proof for the key role of SlpA in host-cell adherence" - this is a significant over-interpretation. The *slpA* mutant is lacking the dominant surface protein, has a surface with completely different biochemical characteristics and has likely changed expression and localisation of multiple other cell surface proteins, several of which have been implicated in adherence. Ascribing a direct role for SlpA in adherence is not justified by these data.

Agree, the SlpA deletion induced changes on the cell surface should be the direct factors affecting the adherence. Then, we changed the description to “indicating the important role of SlpA in host-cell adherence of *C. difficile*” (line 143-144), and discussed more studies are needed in cell surface structures changes due to *slpA* deletion to reveal more targets involved in pathogen-host interaction (line 211-214).

Line 152: 11 is the wrong reference again.

We corrected it to the direct reference. Thanks!

Line 153: "has not been characterized [sic] well" - this is simply not true. The previously described S-layer null mutant that you are referring to here has been extensively characterised in at least two papers (Kirk et al 2017 and Ormsby et al 2023), revealing many of the same phenotypes that you have described here, in addition to several others that you have not tested.

To avoid misunderstanding the truth, we modified the sentence to “However, the insertion mutant could not remove *slpA* completely, and more details about roles of SlpA in *C. difficile* pathogenesis need to be uncovered.” in line 153-154.

Lines 159-160: No, Kirk et al 2017 already demonstrated that *slpA* is not essential and this was subsequently supported by CRISPRi by Müh et al in 2019.

We added more details about the insertion and knockdown mutants and modified the description as “The successful generation of *slpA* deletion mutant further negated the previous knowledge which considered *slpA* essential. Similar with the first S-layer-null insertion mutant of *slpA* (FM2.5) (11) and the CRISPRi knockdown of *slpA* in R20291 (12), deletion of *slpA* resulted in more transparent colony morphology, inhibited growth (Fig. 2A), attenuated toxin production (Fig. 2C and Supplementary Fig. S2B), and suppressed sporulation (Fig. 2D) of *C. difficile* 630Δ*erm*Δ*slpA*, verifying the important roles of SlpA in life cycle of *C. difficile*. Moreover, major pathogenic factors such as interaction with host cells and biofilm formation of the *slpA* deletion mutant were also illustrated in this study and discussed below.” (line 160-167)

Line 161: "growth of which was not obviously affected" - not correct. Kirk et al noted that the *slpA* frameshift mutant reached a maximum stationary phase OD of 2.2 compared with 3.2 for the wild type strain. This is essentially identical to the 60% defect you describe here.

Thanks for your careful check. As above, we discussed the similar results in line 161-167 as “Similar with the first S-layer-null insertion mutant of *slpA* (FM2.5) (11) and the CRISPRi knockdown of *slpA* in R20291 (12), deletion of *slpA* resulted in more transparent colony

morphology, inhibited growth (Fig. 2A), attenuated toxin production (Fig. 2C and Supplementary Fig. S2B), and suppressed sporulation (Fig. 2D) of *C. difficile* 630 Δ *erm* Δ *slpA*, verifying the important roles of SlpA in life cycle of *C. difficile*. Moreover, major pathogenic factors such as interaction with host cells and biofilm formation of the *slpA* deletion mutant were also illustrated in this study and discussed below.”. Thank you!

Line 178-180: If the 42 kDa band were indeed the C terminal fragment of CwpV then a similarly intense band for the larger N terminal fragment would also be apparent.

Thanks! It makes sense. Considering potential loss of some protein bands during cell wall proteins extraction, and the protein size of this band pointed us to CwpV N-terminal domain, we changed our description to “While further studies will be needed for identification of this band to verify the involvement of CwpV.” (line 184-185)

Line 197: the difference is length fairly small and I would not refer to these as filamentous.

Thanks for your good suggestions, we removed filamentous, and changed the sentence to “Disruption of the *slpA* gene resulted in shorter thin cells (Supplementary Fig. S1B) and increased biofilm production (Fig. 4) compared to the wild type...” (line 201)

Line 202-203: the reference to the *D. radiodurans* SlpA here is a little odd as the proteins have absolutely no similarity.

Agree, and we replaced this reference with a recently published close one to emphasize the important roles of S-layer protein in biofilm formation. In the revised version, we changed the description to “and incubation of *C. difficile* with anti-S-layer monoclonal antibodies significantly reduced biofilm formation (25)” in line 206-207. Thanks!

Line 213: again has not been considered essential for many years.

Thanks, and we modified this to “further providing solid proof against the essential role of *slpA* in *C. difficile*.” (line 219-220)

Line 238: References 27 and 28 are not the appropriate original sources for this method.

Thanks for your careful checking. We performed the conjugation based on our previous protocol (ref 27) and modified it with the optimized heat treatment to improve efficiency (we added the very original ref for this as 29).

Lines 257-258: "subcultured in... 10 continuously cycles" - I don't follow what you've done here. Can you please rephrase?

Thanks for pointing out your confusion. To cure the plasmid (harboring antibiotic resistant markers) from the mutants, we cultivated mutants in BHIS medium without antibiotics, and transfer cultures into fresh BHIS medium after 8 h cultivation as a subculture cycle, after repeating 10 subculture cycles, the cultures were spread onto BHIS agar plates with and without antibiotics to check the curing of plasmid.

To reduce confusion, we rephrased the sentence as “Colonies demonstrating only the mutant band were picked and cultivated in BHIS medium without antibiotics. Subcultures were carried

out every 8 h for 10 cycles, followed by being spread onto BHIS to obtain single colonies, which were then replica plated onto BHIS agar and BHIS agar with Tm plates.” (263-265)

Line 266 onwards: Amirkamali is not the original source for this method.

We followed the modified protocol from Amirkamali et al. To acknowledge the original reference, we also cited the original reference from Calabi et al. (line 273) Thanks!

Line 290: 65C for how long?

We added 30 min which we used for heat treatment (line 297). Thanks!

Line 294: How were these inoculated? Very important as that can have a dramatic effect on MIC. "Growth... monitored" - only endpoints are shown so presumably all of these were in stationary phase?

For inoculation: fresh log phase cultures were used for inoculation, and each 96-well contains an initial culture of $\sim 10^5$ CFU.

We used OD absorbance at 10 h cultivation (early stationary phase) to compare their sensitivity to vancomycin. We added this detail as “and growth after 10 h cultivation was used to compare their sensitivity to vancomycin”. (line 305)

Line 337: again I would argue that this is not autolysis.

We changed it to “Triton X-100-induced autolysis”. Thanks!

Figures and legends

Number and type of replicates should be added to all legends.

Number and replicates were added when applicable. Thanks!

Figure 1: As noted above this is likely not Cwp84.

Cwp84 related information was removed. Thanks!

Figure 2: Very little detail in the legend. More here would save the reader referring back and forth to the methods and would make the figure much more understandable.

Thanks for your good suggestions. More details were added into the figure legend.

Figure 3: very little detail here. What does the dotted line in B represent? Also the measurement here is presumably max growth expressed as a percentage of the control. Survival is not correct when talking about MIC.

Thanks. More details were added, and OD reading instead of survival was provided for Fig. 3B.

Figure 4: B has no error bars - single measurements? On rereading C I can't figure out if this is actually representing intensity across a Z stack or not. The legend needs a lot more information to aid interpretation.

Thanks for the good questions. Fig. 4B provides qualitative support to biofilm health using a biofilm test kit. *C. difficile* is a corrosive biofilm. Its biofilm harvests electrons from the stainless-steel coupon underneath it in our test kit. When Fe^0 loses electrons to become Fe^{2+} , corrosion

occurs. More sessile cells harvest more electrons, causing more severe corrosion. Using two separate biofilm test kit units, we compared R_p (corrosion resistance) responses of two different biofilms. The R_p responses support other data indicating better biofilm growth of *C. difficile* 630 Δ *erm* Δ *slpA* strain (showing lower R_p , i.e., higher corrosivity). The R_p responses are reliable qualitative indicator of corrosivity and thus biofilm strength. Regarding error bars for the R_p response, we did not provide them because the R_p sequence in Fig. 4B is consistent with other biofilm data (Fig. 4 A and C). It is common to skip replicates in using qualitative corrosion data used to support core data, after all, the R_p are meant to be qualitative, rather than the quantitative data which require error bars to provide statistical significance.

Like we explained above, CLSM images are used to support biofilm sessile cell information in both 3-D and 2-D modes. In 3-D mode, the biofilms appear to be uneven. Thus, we did not apply z-axis height ratio which is not a good quantitative measurement. Although fluorescence intensity is not very accurate, it is still widely used quantitatively.

Therefore, we modified the figure legend to explain the single measurements and CLSM results.

Figure S1: B is cell length not size presumably. Are these images phase contrast? If so they are extremely over-saturated.

Thank you! We changed cell size to cell length as suggested. The images were taken under brightfield-transmitted light under same conditions but with auto exposure time. The same microscope also allowed us to try phase contrast, but we were not able to get clear images. To improve the images, we slightly decreased the contrasts of both images with the same adjustment degree.

Figure S2: panel A would be much better merged with Fig. 2B. What is shown in S2B? The legend only says toxins [sic] production at 24 h and 48 h cultivation - not nearly enough detail incidentally. Fig 2C apparently also shows "less toxins production" at 24 and 48 h. So it seems these are showing the same thing, yet the values are completely different - there's > an order of magnitude difference in the wt at 48 h. Merging Figure 2B and S2A would help reader understanding.

Thanks for your suggestions! Because Fig. SA and B referred to cell concentration and toxins production, respectively, we think it is helpful to keep them separate for the description.

Appreciate your careful thoughts. Fig. S2B is the intercellular toxins result, and Fig. 2C in the main text is for the extracellular ones. We added more details in the figure legend to avoid the confusion.

Minor typographical issues

Line 35: hospital acquired infection

Line 45: spore forming

Line 54: SLPs are derived

Line 101: loss of

Line 110: *slpA* not *SlpA*

Line 114: toxin production

Line 175, 183 and 185: CWPs

Line 189: *cwp84* italics

Line 221: coli

Line 226: anhydrotetracycline
Lines 308-309: in tissue culture
Line 311: biofilm was washed

We highly appreciate your careful reading and great suggestions, and we have modified the manuscript accordingly! Thank you very much!

Reviewer #2 (Comments for the Author):

The data presented in this paper is compelling, supporting and extending previous observations regarding the contribution of the SlpA on a wide variety of phenotypes including toxin production and the capacity of the strain to sporulate.

We gratefully appreciate your positive comments!

However, to improve the paper, there is a need to contextualise the data presented - how can it be linked to the loss of the S layer and what additional data would need to be included to support these observations. Description of the results is currently very limited and needs to be extended to include a greater explanation of ALL the results presented and what they mean. This could be supported in the discussion with a further linkage to existing literature. If the aim of the paper is to highlight the potential and effective use of CrispR Cas as a mutational system, then generation of an additional mutant in a strain with a different SlpA type is warranted.

Thanks for your constructive suggestions!

To explain and extend our results, we modified the discussion section with:

- 1) We added the comparison with the insertion and CRISPRi knockdown *slpA* mutants about roles of SlpA on phenotypes of *C. difficile*.
- 2) We discussed the changes of other CWPs due to *slpA* deletion and their potential involvement in CDI pathogenesis. And this studying is ongoing in our lab.
- 3) Based on current reports on roles of Cwp84, we discussed *slpA* deletion induced *cwp84* expression change may be related to biofilm formation of *slpA* deleted *C. difficile* mutant.
- 4) We discussed roles of SlpA in pathogen-host interaction based on the adhesion results and indicated that detection of cell wall structures due to *slpA* deletion will help uncovering more targets in CDI pathogenesis.

Thanks for your valuable suggestions! Although we deleted the *slpA* gene for the first time with the CRISPR-Cas9 system, we are not focused on this system in this manuscript.

A review of the written English is required throughout as there are numerous spelling mistakes and a number of poorly written passages.

Thanks for your careful check! The manuscript has been double-checked, and the typos and grammar errors we found have been corrected.

Re: Spectrum04005-23R1 (Revealing roles of S-layer protein (SlpA) in *Clostridioides difficile* pathogenicity by generating the first *slpA* gene deletion mutant)

Dear Dr. Shaohua Wang:

Your manuscript has been accepted, and I am forwarding it to the ASM production staff for publication. Your paper will first be checked to make sure all elements meet the technical requirements. ASM staff will contact you if anything needs to be revised before copyediting and production can begin. Otherwise, you will be notified when your proofs are ready to be viewed.

Sincerely,
Meera Unnikrishnan
Editor
Microbiology Spectrum

Reviewer #1 (Comments for the Author):

This manuscript is a resubmission of a paper that I reviewed earlier this year. When I first read this manuscript, I had concerns about the connection to previous work in the field. The authors have done an admirable job of addressing these concerns and I believe the current version is significantly improved. Overall, I can identify no significant issues in this version and I think this paper will be of great interest in the field.